# Machine-supported decision-making to improve agricultural training participation and gender inclusivity

**Norman Peter Reeves**[1]*, **Ahmed Ramadan**[1,2], **Victor Giancarlo Sal y Rosas Celi**[3], **John William Medendorp**[4], **Harun Ar-Rashid**[5], **Timothy Joseph Krupnik**[6], **Anne Namatsi Lutomia**[7], **Julia Maria Bello-Bravo**[7], **Barry Robert Pittendrigh**[4]

1 Sumaq Life LLC, East Lansing, Michigan, United States of America, 2 Department of Biomedical Engineering, University of Minnesota, Minneapolis, Minnesota, United States of America, 3 Sección de Matemáticas, Departamento de Ciencias, Pontificia Universidad Católica del Perú, Lima, Perú, 4 The Urban Center, Department of Entomology, Purdue University, West Lafayette, Indiana, United States of America, 5 Agricultural Advisory Society (AAS), Dhaka, Bangladesh, 6 Sustainable Agrifood Systems Program, International Center for the Improvement of Wheat and Maize (CIMMYT), Dhaka, Bangladesh, 7 Department of Agricultural Sciences Education and Communication, Purdue University, West Lafayette, Indiana, United States of America

* reevesn@icloud.com

**Data Availability Statement:** All data files are available from https://purr.purdue.edu/publications/3983/1.

## Abstract

Women comprise a significant portion of the agricultural workforce in developing countries but are often less likely to attend government sponsored training events. The objective of this study was to assess the feasibility of using machine-supported decision-making to increase overall training turnout while enhancing gender inclusivity. Using data obtained from 1,067 agricultural extension training events in Bangladesh (130,690 farmers), models were created to assess gender-based training patterns (e.g., preferences and availability for training). Using these models, simulations were performed to predict the top (most attended) training events for increasing total attendance (male and female combined) and female attendance, based on gender of the trainer, and when and where training took place. By selecting a mixture of the top training events for total attendance and female attendance, simulations indicate that total and female attendance can be concurrently increased. However, strongly emphasizing female participation can have negative consequences by reducing overall turnout, thus creating an ethical dilemma for policy makers. In addition to balancing the need for increasing overall training turnout with increased female representation, a balance between model performance and machine learning is needed. Model performance can be enhanced by reducing training variety to a few of the top training events. But given that models are early in development, more training variety is recommended to provide a larger solution space to find more optimal solutions that will lead to better future performance. Simulations show that selecting the top 25 training events for total attendance and the top 25 training events for female attendance can increase female participation by over 82% while at the same time increasing total turnout by 14%. In conclusion, this study supports the use of machine-supported decision-making when developing gender inclusivity policies in agriculture extension services and lays the foundation for future applications of machine learning in this area.

**Funding:** This publication was made possible in part by internal funds from Michigan State University [JBB & BRP]. The fall armyworm animation and scaled extension activities were supported by funds provided to CIMMYT by the Borlaug Higher Education for Agricultural Research and Development Program (BHEARD; USAID award # AID-BFS-G-11-00002) at Michigan State University [JWM] and the Bill and Melinda Gates Foundation support for the Cereal Systems Initiative for South Asia (CSISA; USAID award #BFS-IO-17-00005 and BMGF isINV-009787) [TJK]. This work was also supported by the One CGIAR Regional Integrated Initiative Transforming Agrifood Systems in South Asia (TAFSSA) [TJK]. USAID: https://www.usaid.gov BMGF: https://www.gatesfoundation.org TAFSSA: https://www.cgiar.org/initiative/20-transforming-agrifood-systems-in-south-asia-tafssa Its contents are solely the responsibility of the authors and do not necessarily represent the official views of Michigan State University, CIMMYT, BHEARD, USAID, the Bill and Melinda Gates Foundation, CSISA or TAFSSA. No sponsors or funders played any role in the study design, data collection and analysis, decision to publish, or preparation of the manuscript.

**Competing interests:** I have read the journal's policy and the authors of this manuscript have the following competing interests: N. Peter Reeves is the Founder and President of Sumaq Life LLC. Ahmed Ramadan is a part-time employee of Sumaq Life LLC. Sumaq Life LLC applies mathematical modeling approaches to understand complex systems in order to optimize their performance. It receives funding for these services, including work on the current project. This does not alter our adherence to PLOS ONE policies on sharing data and materials. The remaining authors declare no competing interest in the production of this work".

## Introduction

Improving agricultural practices is critically important for developing countries. Although the agriculture sector only accounts for 4% of global gross domestic product (GDP), it's impact on developing countries is more substantial, often accounting for more than 25% of national GDP [1]. Approximately 65% of workers in developing countries are employed in the agricultural sector [1], with 43% of those being women [2]. Given the strong link between agricultural output and the health of a country's economy [3, 4] and the health of its citizens [5], many governments are undertaking large-scale training programs to improve agricultural practices [6, 7].

There is a growing body of scientific knowledge to improve agricultural practices [8]; however, translating scientific insights to end-users, often low-literacy and non-English speaking individuals, is challenging [9]. To scale agricultural training, government extension services have begun to deploy information and communication technologies (ICTs) as part of their agricultural programs [10, 11]. Examples of scalable ICTs include linguistically localized, computer-animated training videos that are specifically developed for low-literacy, non-English speaking populations [12, 13]. Such interventions have been shown to cause learning gains and adoption of technologies taught in the animations, as well as innovations in participating communities post-intervention [14–16].

While ICTs increase access to agricultural extension services for production [17, 18], their deployment likely would be enhanced by taking into account gender-based training patterns (e.g., preferences and availability for training). Recent analysis of ICT-delivered agricultural training indicates that differences in male and female farmers exists [19]. This study as well as others have shown that male and female farmers may differ in terms of when and where they prefer training to be conducted and whether training is male- or female-led [20–24]. As such, choosing times in the day, days of the week, locations and venues, and gender of the trainer to promote female participation may thwart efforts to increase overall attendance at government sponsored training events [19]. Therefore, designing effective agricultural training programs that may include multiple objectives, (e.g., increasing general attendance while maintaining a certain proportion of female participation), may not be trivial.

More formal mathematical approaches for selecting times, locations and venue types, and gender to conduct training may help improve agricultural training participation and gender inclusivity. Although there is considerable research into the use of machine learning in agriculture, the applications mainly focus on crop, livestock, water and soil management [25]; there does not appear to be applications using machine learning to improve agricultural extension participation, including promoting gender inclusivity. We hypothesize that machine-supported decision-making that accounts for gender-based training patterns can concurrently increase total and female participation over that of the current approach that relies only on human intuition. If results support this hypothesis, incorporating machine learning as part of ICTs, and other modes of information dissemination, in large-scale agricultural extension programs is warranted.

## Materials and methods

This observational study used existing data recorded at agricultural extension events conducted throughout Bangladesh. Data was collected by the Agricultural Advisory Society (AAS) in partnership with the International Center for the Improvement of Wheat and Maize (CIMMYT). Data were not specifically collected for research purposes, but instead to determine the overall level of participation in extension training. Participation information reflected aggregated data and as such was recorded in a manner that the identities of individuals could not be ascertained. This study was deemed exempt by Michigan State University Biomedical and

Health Institutional Review Board (IRB 00004626). Obtaining informed consent was not possible given that data were de-identified.

## Data collection

In total, 1,080 AAS/CIMMYT extension training events were held between October 2018 and January 2019 across four Bangladesh divisions: Rangpur, Khulna, Dhaka, and Rajshahi. During AAS/CIMMYT extension training events, one or more of the following ICT-delivered agricultural training videos were shown: (1) a video for mitigating Fall Armyworm [26], a major invasive pest resulting in significant crop loss [27, 28], (2) a video for planting healthy rice seedlings [29], and (3) a video promoting earlier dates for planting to yield more wheat, the second most important food crop in Bangladesh [30]. At each training event, three trained enumerators collected data on the total number of persons attending, the number of females attending, the gender of the extension agents delivering the training, time of the day, day of the week, month of the year, and location and type of venue at which videos were shown. Training locations and venue types for video presentations were purposefully selected to maximize attendance in order to disseminate information on these topics in advance of the primary maize, rice, and wheat cropping seasons. No incentives were provided to participate (e.g., no meals, no payments). Most events were not pre-organized to recruit participants, but rather involved setting up the video show and attracting a crowd.

## Machine learning

Model outputs for the two gender-specific training models were (1) total number of males and (2) total number of females attending training events. In a recent study, statistical models were used to study factors that could be associated with total (males + females) and female average attendance only, and to estimate the strength of these associations [19]. Using those results as starting points in addition to first order interactions, model inputs for the two gender specific training models, representing situational factors under the control of designers of extension training programs, included the following: gender of the trainer, time of the day, day of the week, type of venue, location in Bangladesh (specifically the division hosting the event) and first order interactions of division with time of the day, type of venue, and gender. Time of the day was parsed into before 11:00, 11:00–15:30, and after 15:30. Venue types included educational institutions, farmers' houses, marketplaces, religious institutions, shops, tea stalls, union parishad campuses (governmental council building grounds) and "other" venues (open spaces in front of hospitals, sport clubs, rail gates, bus stands, playgrounds). The "other" category was created to combine training events with fewer than nine observations for a given venue type. Month of the year was studied to assess if the two gender specific training models changed with time.

To develop more parsimonious models to improve precision in model predictions, the number of model inputs were reduced for the two gender-specific training models using a shrinkage approach. Three different shrinkage approaches were considered: group Lasso (group Lasso), group Minimax Convex Penalty (group MCP), and group Smoothly Clipped Absolute Deviation (group SCAD). Group selection was chosen since several model inputs were grouped by factors (time of the day, day of the week, month of the year, venue type, and Bangladesh division). All shrinkage approaches used datasets for learning (randomly selected 80% of data) and testing (remaining 20% of data) to train and evaluate the gender-specific training models. All three model shrinkage approaches used cross-validation to select the optimal shrinkage parameter [31]. The smallest standardized mean squared error (SMSE) in the testing dataset was used to choose the shrinkage approach, and therefore, to select the most

influential model inputs (details are presented in the appendix). This process was implemented in the statistical software R [32].

Next, using the most influential model inputs that were derived for the male and female models respectively, generalized linear models were developed to capture gender-based training patterns. These models used a Poisson distribution with a dispersion parameter for male and female attendance. Modeling was done in MATLAB using the function 'fitglm' to estimate the model parameters, mapping model inputs into the model output space.

With the two gender-specific training models defined, the expected attendance was estimated, given the set of training choices. For this purpose, a mesh was created in the discrete space with each model input variable representing an axis. Model inputs eliminated during model shrinkage were removed from the mesh. The mesh was used to represent the various possible training events that could occur (e.g., female-led training event held in a farmer's house in Rangpur before 11:00 am). At each point, total attendance and female attendance were predicted (i.e., point estimates) using the male and female models. Predictions were performed in MATLAB using the function 'predict'.

To estimate the variability in the data and to assess differences in attendance between various training events, a bootstrapping approach was applied. Bootstrapping involved creating 1,000 new datasets by sampling with replacement from the original dataset. For each dataset, new male and female models were created and point estimates calculated. Combining the point estimates from all 1,000 datasets, the average total attendance, female attendance, and proportion of female attendance were obtained with 95% confidence intervals. Using the average predicted attendance, two lists were created, one representing the top 100 training events (i.e., mesh points) for total attendance and the other the top 100 training events for female attendance. To clarify, the use of the term "top training events" used hereafter refers to those training events with the highest attendance, be it total or female attendance, based on the gender of the trainer, and when and where training was conducted; "top training events" do not reflect the quality of training or some other attribute.

## Model simulations

An algorithm was created to simulate the effects of weighting the importance of female participation over total attendance. This was done by selecting the top X% from the female list and replacing the lower X% from the total list, where X% represents a range (0%-100% with 10% increments). For example, the top 50%, reflecting a balance between increasing total attendance and increasing female participation, would take the top 50 training events from the total list and the top 50 training events from the female list to create a new list of 100. The new list would then be used to estimate the average total attendance, female attendance, and proportion of female attendance. This was done to qualitatively capture the effects of weighting female attendance over total attendance to help develop sensible gender inclusivity goals.

Reduced training variety, representing smaller lists of the top training events, is expected to improve model performance and lead to greater total attendance, female attendance, and proportion of female attendance; however, reducing training variety may also limit future model performance. Because the models are still in their infancy, future iterations with more observations are expected to improve machine learning and subsequent model performance. Greater training variety provides a larger solution space and increases the odds of finding the optimal solution. To explore the trade-off between model performance and machine learning, simulations were performed with lists of the top 100, top 50, and top 10 training events for total and female attendance. This was done to qualitatively capture the effects of training variety on total attendance, female attendance, and the proportion of female attendance.

### Evaluated hypothesis

The objective of this study is to provide evidence that machine-supported decision-making for selecting training events can improve the current approach, based on human intuition. As a benchmark, the average total attendance and average female attendance across all past extension training events will be used. If the machine learning approach significantly exceeds these averages, it provides strong justification for the addition of machine-supported decision-making in extension services.

### Inclusivity in global research

Additional information regarding the ethical, cultural, and scientific considerations specific to inclusivity in global research is included in the Supporting Information.

## Results

In total, there were 132,358 Bangladeshi farmers who attended the 1,080 training events; however, there was missing information on 13 (1.2%) of the training sessions, which were removed from the analysis. Based on t-tests, there was no evidence of differences in the average number of male or female attendance between the included and not included training sessions. Among the included 1,067 training events (130,690 farmers) in the four Bangladesh divisions, 527 events (49.4%) occurred in Rangpur, 295 events (27.6%) in Khulna, 202 events (18.9%) in Dhaka, and only 43 events (4.0%) in Rajshahi (Table 1). The average total attendance was 122.0 (SD = 103.6, Range = 15–600) with an average female attendance of 23.0 (SD = 25.8, Range = 0–150). These averages represent the benchmarks that the machine learning approach will be compared against.

In terms of model shrinkage approaches, group Lasso had the lowest accuracy, assessed using standardized mean square error, in predicting attendance in the test dataset (Table 2). Most of the variables selected were common between the three approaches, although group Lasso tended to select more variables than the other two shrinkage approaches. Based on the accuracy and number of variables selected, the group MCP approach was used to shrink the male and female model. Subsequent analysis used the model inputs selected with this approach. The shrinked male model included an intercept, gender of trainer, venue type, and the interaction with all lower-order terms of Bangladesh division and time of the day. The shrinked female model included an intercept, gender of trainer, time of the day, month of the year, venue type, and Bangladesh division.

Using the shrinked male and female models, the top 100 training events were predicted for total attendance and female attendance (Tables 3 and 4 show the top 10 from this list). As shown in Tables 3 and 5, model-predicted attendance is in agreement with observed attendance, suggesting cautiously the models reflect the real world. As can be seen in Tables 3 and 4, most of the predicted top training events to improve total attendance were male-led (9 out of 10); in contrast, most of the top training events to improve female attendance were female-led (8 out of 10), highlighting a clear preference for females to receive educational information from female trainers. Also, most of the predicted top training events for total attendance were held after 15:30 (9 out of 10) in marketplaces (5 out of 10), while females were predicted to attend either training before 11:00 (4 out of 10) or after 15:30 (5 out of 10) and had a preference for training to be held at farmers' houses (7 out of 10), particularly in the province of Rajshahi (8 out of 10).

On average, selecting more training events from the top female list led to less total attendance, and not surprisingly, increased female attendance and the proportion of female attendance at training events (Fig 1). Also, reducing training variability from lists of 100 to 50 to 10

**Table 1. Descriptive statistics of the 1,067 agricultural training events conducted in Bangladesh between October 2018 and January 2019.**

| Characteristics | N (%) | Number of trained Males | Number of trained Females |
|---|---|---|---|
| | | Mean (Standard deviation, Range) | Mean (Standard deviation, Range) |
| **Gender** of Trainer | | | |
| Female | 104 (9.7) | 36 (50, 4–280) | 29.7 (16.9, 0–120) |
| Male | 963 (90.3) | 106 (104, 4–600) | 22.5 (26.5, 0–150) |
| **Time** of the day | | | |
| before 11:00 | 307 (28.8) | 58.9 (60.1, 4–270) | 32.2 (26.7, 0–150) |
| 11:00–15:30 | 238 (22.3) | 37.8 (27.0, 6–175) | 24.6 (21.0, 0–130) |
| after 15:30 | 522 (48.9) | 151.1 (116.2, 6–600) | 17.3 (25.7, 0–150) |
| **Day** of the week | | | |
| Sunday | 154 (14.4) | 112.3 (108.8, 6–500) | 20.0 (24.3, 0–120) |
| Monday | 142 (13.3) | 105.5 (103.7, 6–440) | 24.1 (28.1, 0–100) |
| Tuesday | 137 (12.8) | 90.1 (99.1, 8–400) | 23.2 (25.6, 0–150) |
| Wednesday | 147 (13.8) | 103.5 (110.6, 4–600) | 25.3 (25.1, 0–100) |
| Thursday | 154 (14.4) | 100.1 (101.7, 10–400) | 20.6 (24.1, 0–130) |
| Friday | 166 (15.6) | 89.0 (89.9, 8–420) | 24.4 (27.4, 0–150) |
| Saturday | 167 (15.7) | 95.2 (100.7, 4–450) | 24.9 (25.5, 0–130) |
| **Month** of the year | | | |
| October 2018 | 134 (12.6) | 83.9 (77.8, 6–399) | 12.8 (15.4, 0–70) |
| November 2018 | 299 (28.0) | 78.7 (81.4, 4–420) | 19.6 (17.5, 0–100) |
| December 2018 | 391 (36.6) | 109.4 (112.2, 6–450) | 23.8 (24.8, 0–130) |
| January 2019 | 243 (22.8) | 116.8 (114.1, 9–600) | 32.4 (35.6, 0–150) |
| **Venue** type | | | |
| Educational institutions | 54 (5.1) | 140.4 (121.5, 13–450) | 15.6 (23.2, 0–120) |
| Farmers' houses | 460 (43.1) | 54.9 (61.4, 4–400) | 37.6 (26.7, 0–150) |
| Marketplaces | 279 (26.1) | 173.6 (122.3, 18–600) | 5.5 (10.4, 0–60) |
| Religious institutions | 30 (2.8) | 43.8 (47.2, 13–250) | 26.9 (16.4, 4–80) |
| Shops | 41 (3.8) | 73.7 (66.0, 9–300) | 20.8 (23.6, 0–80) |
| Tea stalls | 98(9.2) | 81.2 (69.5, 15–300) | 22.2 (22.4, 0–100) |
| Union parishad campuses | 19 (1.8) | 151.3 (129.0, 20–420) | 7.4 (15.5, 0–50) |
| Other venues | 18 (1.7) | 124.9 (97.5, 8–330) | 20.1 (27.5, 0–100) |
| **Division** of Bangladesh | | | |
| Rangpur | 527 (49.4) | 118.2 (119.3, 6–600) | 27.6 (30.2, 0–150) |
| Khulna | 295 (27.6) | 81.1 (81.8, 4–300) | 21.0 (19.8, 0–100) |
| Dhaka | 202 (18.9) | 80.0 (74.7, 6–399) | 14.0 (14.1, 0–61) |
| Rajshahi | 43 (4.0) | 83.0 (62.7, 9–300) | 27.7 (32.2, 0–130) |

**Table 2. Accuracy (standardized mean square error, SMSE) and number of model inputs selected in the testing dataset for the male and female specific training models using the three model shrinkage approaches.**

| | group Lasso | | group MCP | | group SCAD | |
|---|---|---|---|---|---|---|
| | Accuracy (SMSE) | Number of Variables | Accuracy (SMSE) | Number of Variables | Accuracy (SMSE) | Number of Variables |
| Male training model | 68.72 | 6[a] | 67.98 | 4[b] | 67.97 | 4[c] |
| Female training model | 14.85 | 6[b] | 14.55 | 5 | 14.69 | 3 |

[a] Plus three interactions,

[b] plus one interaction,

[c] plus two interactions.

**Table 3. Predicted top 10 most attended training events for total attendance.**

| rank | Hour | Trainer Gender | Venue | Hub | Predicted Total Attendance[a] | Observed Total Attendance[b] | # of observations[c] |
|---|---|---|---|---|---|---|---|
| 1 | after 15:30 | Male | Marketplaces | Rangpur | 226 (209–244) | 233 (37–660) | (130,54) |
| 2 | after 15:30 | Male | Marketplaces | Khulna | 207 (182–235) | 185 (48–300) | (32,0) |
| 3 | after 15:30 | Male | Union Parishad Campuses | Rangpur | 206 (156–274) | 229 (64–470) | (11,5) |
| 4 | after 15:30 | Male | Educational Institutions | Rangpur | 193 (162–231) | 206 (13–550) | (26,16) |
| 5 | after 15:30 | Male | Union Parishad Campuses | Khulna | 189 (140–257) | Not observed | (0,0) |
| 6 | before 11:00 | Male | Marketplaces | Rangpur | 179 (152–212) | 83 (36–153) | (8,5) |
| 7 | after 15:30 | Male | Tea-stall | Rangpur | 178 (147–216) | 81 (49–130) | (8,1) |
| 8 | after 15:30 | Male | Educational Institutions | Khulna | 176 (144–215) | 160 (47–300) | (9,0) |
| 9 | after 15:30 | Male | Marketplaces | Rajshahi | 169 (123–233) | 158 (33–300) | (7,0) |
| 10 | after 15:30 | Female | Marketplaces | Rangpur | 168 (130–218) | 280 (280–280) | (1,0) |

[a] Average (lower and upper limits of 95% confidence interval).

[b] Average (minimum and maximum of observed values).

[c] Number of observations for males and females, respectively

**Table 4. Predicted top 10 most attended training events for female attendance.** Note that the top female training events occurred in the month of January where there was little observed data.

| rank | Hour | Trainer Gender | Venue | Hub | Predicted Female Attendance[a] | Observed Female Attendance[b] | # of observations |
|---|---|---|---|---|---|---|---|
| 1 | after 15:30 | Female | Farmers' Houses | Rajshahi | 96 (68–137) | Not observed | 0 |
| 2 | before 11:00 | Female | Farmers' Houses | Rajshahi | 88 (62–124) | Not observed | 0 |
| 3 | after 15:30 | Male | Farmers' Houses | Rajshahi | 80 (59–109) | Not observed | 0 |
| 4 | before 11:00 | Male | Farmers' Houses | Rajshahi | 73 (54–99) | Not observed | 0 |
| 5 | after 15:30 | Female | Farmers' Houses | Rangpur | 72 (58–91) | Not observed | 0 |
| 6 | after 15:30 | Female | Tea-stall | Rajshahi | 66 (45–98) | Not observed | 0 |
| 7 | before 11:00 | Female | Farmers' Houses | Rangpur | 66 (53–81) | 50 (50–50) | 1 |
| 8 | 11:00–15:30 | Female | Farmers' Houses | Rajshahi | 65 (46–91) | Not observed | 0 |
| 9 | after 15:30 | Female | Religious Institutions | Rajshahi | 61 (39–97) | Not observed | 0 |
| 10 | before 11:00 | Female | Tea-stall | Rajshahi | 60 (41–89) | Not observed | 0 |

[a] Average (lower and upper limits of 95% confidence interval).

[b] Average (minimum and maximum of observed values).

increased total attendance, but had mixed results in female attendance and the proportion of female attendance. Smaller lists negatively impacted female attendance when selecting solely from the top total training events, but improved female attendance and proportion of female attendance when selecting more from the top female training events.

## Discussion

The objective of this study was to determine if machine-supported decision-making could improve training attendance and gender inclusivity, which would then justify its use in designing large-scale agricultural training programs. The results suggest that, if gender inclusivity goals are sensibly designed using machine-supported decision-making, total attendance and female participation can be concurrently increased. The results, however, also show that overly emphasizing female participation can have negative consequences for total training turnout.

**Table 5. Model predicted versus observed female attendance.**

| rank | Hour | Trainer Gender | Venue | Hub | Predicted Female Attendance[a] | Observed Female Attendance[b] | # of observations |
|---|---|---|---|---|---|---|---|
| 7 | before 11:00 | Female | Farmers' Houses | Rangpur | 66 (53–81) | 50 (50–50) | 1 |
| 11 | after 15:30 | Male | Farmers' Houses | Rangpur | 60 (52–68) | 63 (0–150) | 33 |
| 16 | before 11:00 | Male | Farmers' Houses | Rangpur | 54 (48–61) | 60 (9–150) | 46 |
| 26 | 11:00–15:30 | Female | Farmers' Houses | Rangpur | 48 (39–60) | 47 (38–55) | 2 |
| 36 | after 15:30 | Male | Tea-stall | Rangpur | 41 (33–52) | 30 (30–30) | 1 |
| 41 | 11:00–15:30 | Male | Farmers' Houses | Rangpur | 40 (35–46) | 35 (8–100) | 37 |
| 43 | after 15:30 | Male | Religious Institutions | Rangpur | 38 (28–52) | 28 (14–41) | 2 |
| 46 | before 11:00 | Male | Tea-stall | Rangpur | 37 (29–48) | 36 (25–46) | 2 |
| 53 | before 11:00 | Male | Religious Institutions | Rangpur | 35 (25–47) | 26 (17–29) | 4 |
| 57 | after 15:30 | Male | Shop | Rangpur | 34 (24–46) | 36 (0–80) | 3 |

[a] Average (lower and upper limits of 95% confidence interval).

[b] Average (minimum and maximum of observed values).

For instance, selecting all the training events from the top female list would increase female attendance and the proportion of female attendance to ~ 0.5 but would decrease total attendance below that of the benchmark average of 122 attendees per event (Fig 1). Moreover, many of the top training events for females are predicted to take place in Rajshahi. Focusing on this one division may be beneficial for overall female attendance, but it would limit gains in female participation to one region and may not fulfill the goals of the extension program. Follow-up simulations show that balancing training across all divisions does not dramatically reduce overall attendance or female participation and still exceeds total and female averages based on human intuition (see S1 File). As shown in this study, machine-supported decision-making could provide important insights by allowing policy makers to first simulate "what if" scenarios before applying policies in the real-world.

Many of the top female training sessions were not observed, suggesting cautiously (as it is outside of the observed range of data) that machine learning could be useful for uncovering novel training events to improve female attendance. The data also highlights the need to balance model performance with machine learning: over emphasizing early model performance to increase attendance may restrict machine learning, which could then limit model future performance.

The inclusion of the variable month in the female model indicated that female attendance increased over time, suggesting that the female model had time-varying properties. This is an interesting observation that suggests females' willingness to attend training increased by the end of the study period, although it is also possible that improved methods for female outreach, or some other causes, could explain the phenomena.

Of pragmatic importance, the gender-based training models provide important insights into those training events that enhance participation. In general, training events that were male-led and held at marketplaces after 15:30 tended to improve total training attendance while training events that were female-led and held at farmers' houses, either before 11:00 or after 15:30, tended to increase female attendance. Also, model predictions were more likely to fall within the observed data range when there were more observations (Tables 3 and 5), highlighting the need for sufficient machine learning in early model development. However, both male and female models appeared to make reasonable predictions that matched the observed data. Note that the female model predicted many top training events that were not observed (Table 4). This is because the female model included the variable

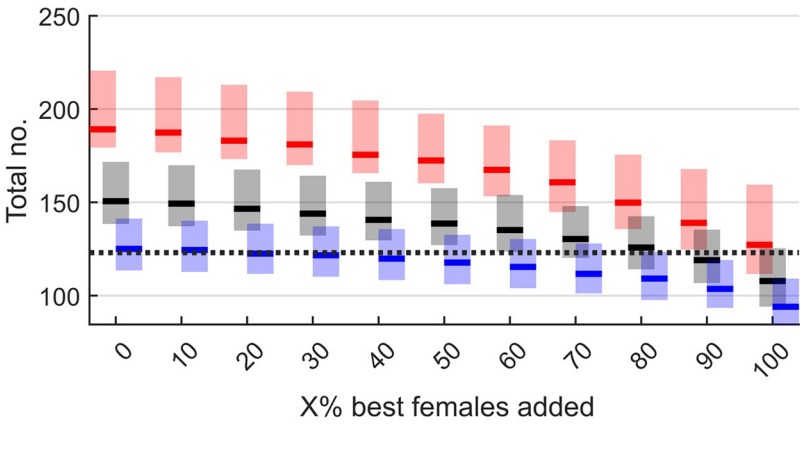

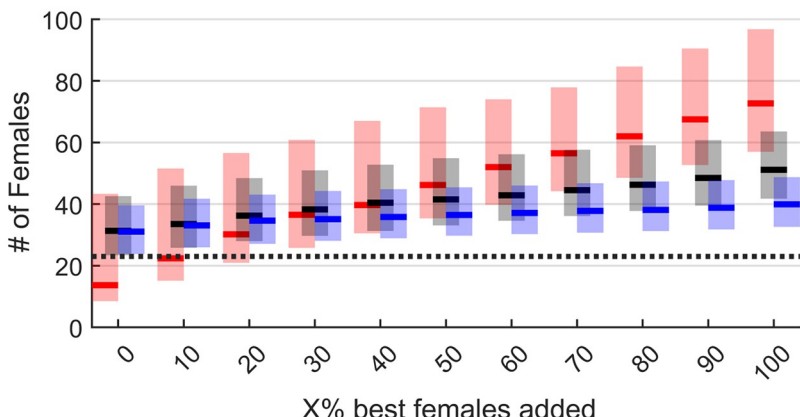

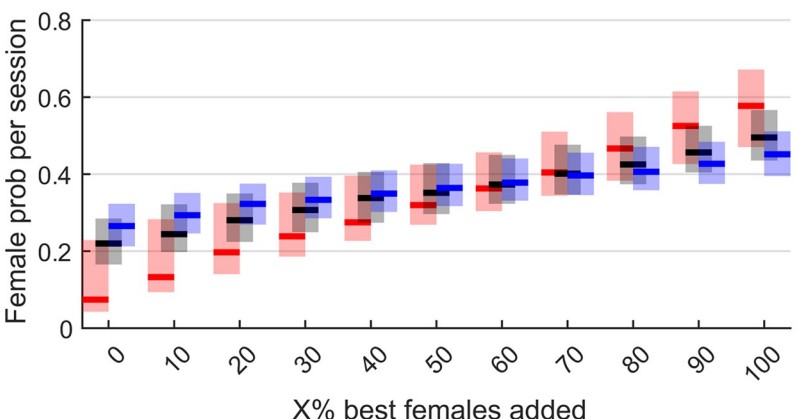

**Fig 1. Bootstrapping results for average total attendance (top), average female attendance (middle), and proportion of female attendance (bottom).** Blue represents the top (i.e., most attended) training lists of 100, black represents lists of the top 50, and red represents lists of the top 10. Shaded boxes represent the 95% confidence intervals. As a benchmark, the thick dotted line represents the average total attendance (122 persons per session) and female attendance (23 females per session) when selecting training events using human intuition. The goal is for the lower 95% limits to exceed these averages, thus suggesting that machine-supported decision-making can help improve training attendance and gender inclusivity.

month as an effect modifier, which restricted the observation to the month of January, thus reducing the number of observations.

The study presents some limitations. Given the short period of data collection, the ability to understand time-varying trends in the data is limited. These could be periodic fluctuations related to seasonal changes, including cyclic planting seasons, but also non-periodic fluctuations, such as gradual cultural acceptance of female farmers. In addition, the observational nature of the study could limit the generalizability of the study findings. Being observational, the study was not designed to ensure sufficient sampling across all variables to allow for an accurate assessment of their impact on training attendance. Note that only 4% of all training sessions were completed in Rajshahi and a significant percentage of training events (23.8%) had zero female participation. Components of the model with more observations normally improve confidence in modeling predictions (i.e., smaller confidence intervals) and, therefore, bias the selection of model inputs to those with more observations. Finally, this observational study likely missed other important factors impacting training turnout. Future data collection should consider investigating potential underlying influences (cultural and otherwise) impacting gender-based training patterns, such as the workload that prospective participants might face at different times of the day or days of the week, agricultural cropping patterns, and the use of incentives or other recruiting strategies (e.g., female-promoted events).

## Optimizing outcomes

In terms of selecting an optimal model performance versus machine learning trade-off, one solution would be to select the *X% top female added* that significantly exceeded total attendance and female attendance averages in benchmarking (122 and 23 respectively) while allowing for the largest list to be used to promote machine learning. Based on the results, to improve both total attendance and gender inclusivity, a top training list of 50 with 50% of training events coming from the female list is recommended. Although the smaller list of 10 would improve total attendance and female attendance, both male and female models would benefit from increase training variety to improve their future performance. Training variety could be reduced after more data is collected and optimal solutions are found. Based on simulations, maintaining training variety by selecting the top 25 training events for total and female attendance, female participation can be increased by over 82% while at the same time increasing total turnout by 14% (Fig 1).

To facilitate machine learning, continuous sampling and processing of data from agricultural training programs are recommended, i.e., (1) hosting training events, (2) learning computationally from those events, (3) enacting recommendations for the next set of training events, and (4) repeating to optimize outcomes. In subsequent rounds of data collection, the project could shift from being purely observational to using optimal experimental design [33] to focus data collection to help refine models further and to improve their performance. Additionally, if possible, recording changes in farmer output would be valuable to assess the impact of training on farm productivity and could serve as an additional model output used for optimization purposes.

Although still in its infancy, machine-supported decision-making appears to be a viable tool for improving agricultural training attendance and gender inclusivity. By increasing training turnout and gender inclusivity, this machine learning application is expected to enhance the social and economic health of developing countries and the health of their populations. This study, to the authors' knowledge is the first to use machine learning tools around gender inclusivity within the context of scaling ICT interventions in a developing country context.

This work sets the stage for future machine learning efforts that drive rounds of ICT interventions towards optimizing agricultural training programs and their derived benefits.

## Appendix

In order to implement the three different shrinkage approaches with the training dataset, the library *grpreg* in R was used. All three shrinkage approaches relied on the estimation of the regularization parameter λ, used to select the number of groups (i.e., model inputs) to retain. To estimate λ, a 10-fold cross-validation was implemented using the function *cv.grpreg* within the *grpreg* library. More specifically, the dataset was divided in ten groups, and the following procedures were conducted for each group:

1. Fit the penalized regression for the nine groups.

2. Predict cross-validation error with the excluded group using the fitted model.

Both steps were conducted over a grid of values of λ. Therefore, for a given value of λ, the mean and standard deviation of the predicted cross-validation error was computed. The value of λ that minimizes the cross-validation error defined the final model proposed for each technique. To clarify, for each proposed model, the value of λ determines the selected number of

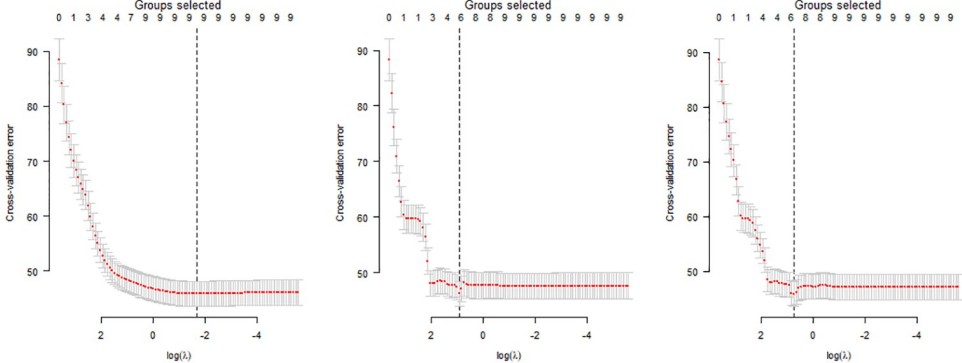

**Fig 2. Cross-validation for model parameter selection for group Lasso (left), group MCP (center) and grouped SCAD (right) considering the total number of male attendees as a model output variable in the training dataset.**

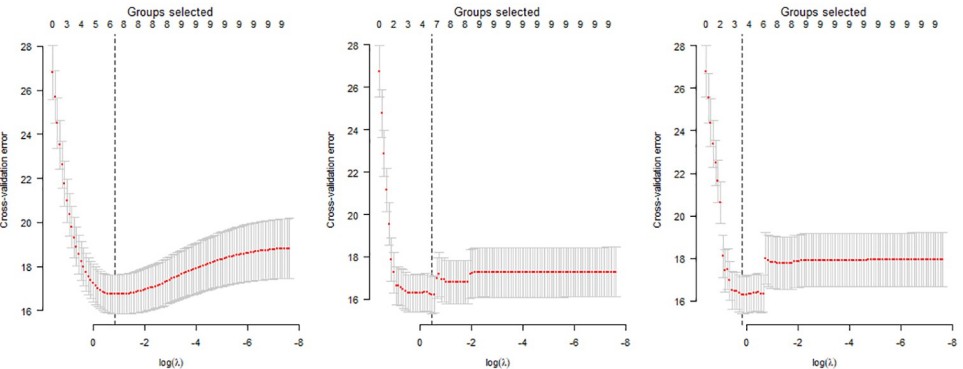

**Fig 3. Cross-validation for model parameter selections for group Lasso (left), group MCP (center) and group SCAD (right) considering the total number of female attendees as a model output variable in the training dataset.**

groups used as model inputs during the parameter selection process. Figs 2 (Male model) and 3 (Female model) show a scatterplot of the predicted cross-validation error over the grid for λ.

## Supporting information

**S1 File.**
(DOCX)

## Author Contributions

**Conceptualization:** Norman Peter Reeves, Ahmed Ramadan, Victor Giancarlo Sal y Rosas Celi, Barry Robert Pittendrigh.

**Data curation:** Norman Peter Reeves, Ahmed Ramadan, Victor Giancarlo Sal y Rosas Celi, Harun Ar-Rashid, Timothy Joseph Krupnik.

**Formal analysis:** Norman Peter Reeves, Ahmed Ramadan, Victor Giancarlo Sal y Rosas Celi.

**Funding acquisition:** John William Medendorp, Timothy Joseph Krupnik, Julia Maria Bello-Bravo, Barry Robert Pittendrigh.

**Investigation:** Harun Ar-Rashid.

**Methodology:** Norman Peter Reeves, Ahmed Ramadan, Victor Giancarlo Sal y Rosas Celi.

**Project administration:** Norman Peter Reeves.

**Supervision:** Barry Robert Pittendrigh.

**Validation:** Norman Peter Reeves, Ahmed Ramadan, Victor Giancarlo Sal y Rosas Celi.

**Visualization:** Norman Peter Reeves, Victor Giancarlo Sal y Rosas Celi.

**Writing – original draft:** Norman Peter Reeves, Ahmed Ramadan, Victor Giancarlo Sal y Rosas Celi.

**Writing – review & editing:** Norman Peter Reeves, Ahmed Ramadan, Victor Giancarlo Sal y Rosas Celi, John William Medendorp, Harun Ar-Rashid, Timothy Joseph Krupnik, Anne Namatsi Lutomia, Julia Maria Bello-Bravo, Barry Robert Pittendrigh.

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
