## [Decision Letter · Decision Letter 0]

14 Aug 2022

PONE-D-22-16835Data-driven Modeling to Improve Agricultural Training Participation and Gender InclusivityPLOS ONE

Dear Dr. Reeves,

Thank you for submitting your manuscript to PLOS ONE. After careful consideration, we feel that it has merit but does not fully meet PLOS ONE’s publication criteria as it currently stands. Therefore, we invite you to submit a revised version of the manuscript that addresses the points raised during the review process.

The material in the paper looks interesting, although the manuscript should be revised carefully to meet PLOS ONE publication criteria. 

Please take particular care to the comments raised by the two reviewers.

The paper presentation should be cared more, in particular it should be provided clearer motivations.

A reviewer criticizes also the cited body of research, which looks not to be exhaustedly covered, and he suggests to enlarge explicitly the "Related Work" section.

You should pay attention to provide more motivation and description on the used methodology, providing a clearer validity about the overall approach.

Explain better and motivate the employed dataset and features, in particular to the data used for the training split.

Please take carefully into account all the comments for improving the manuscript to meet PLOS ONE standards before resubmitting it to the journal.

We look forward to receiving your revised manuscript.

Kind regards,

Sergio Consoli

Academic Editor

PLOS ONE

Journal Requirements:

“I have read the journal's policy and the authors of this manuscript have the following competing interests:

N. Peter Reeves is the Founder and President of Sumaq Life LLC.  Ahmed Ramadan is a part-time employee of Sumaq Life LLC.  Sumaq Life LLC applies mathematical modeling approaches to understand complex systems in order to optimize their performance.  It receives funding for these services, including work on the current project. The remaining authors declare no competing interest in the production of this work.”

“This work was supported by the Borlaug Higher Education for Agricultural Research and Development Program (BHEARD) and USAID under Grant # AID-BFS-G-11-00002 and #BFS IO-17-00005; and the Bill and Melinda Gates Foundation support for the Cereal Systems Initiative for South Asia (CSISA) under Grant # BMGF INV-009787. Its contents are solely the responsibility of the authors and do not necessarily represent the official views of USAID, BHEARD, CSISA, or the Bill and Melinda Gates Foundation.”

“This publication was made possible in part by internal funds from Michigan State University [JBB & BRP]. The fall armyworm animation and scaled extension activities were supported by funds provided to CIMMYT by the Borlaug Higher Education for Agricultural Research and Development Program (BHEARD; USAID award # AID-BFS-G-11-00002) at Michigan State University [JWM] and the Bill and Melinda Gates Foundation support for the Cereal Systems Initiative for South Asia (CSISA; USAID award #BFS-IO-17-00005 and BMGF isINV-009787)[TJK].

USAID: https://www.usaid.gov

BMGF: https://www.gatesfoundation.org

Its contents are solely the responsibility of the authors and do not necessarily represent the official views of USAID, BHEARD, CSISA, or the Bill and Melinda Gates Foundation.

No sponsors or funders played any role in the study design, data collection and analysis, decision to publish, or preparation of the manuscript.”

Reviewers' comments:

Reviewer's Responses to Questions

**Comments to the Author**

1. Is the manuscript technically sound, and do the data support the conclusions?

Reviewer #1: Partly

Reviewer #2: No

2. Has the statistical analysis been performed appropriately and rigorously? 

Reviewer #1: No

Reviewer #2: No

3. Have the authors made all data underlying the findings in their manuscript fully available?

Reviewer #1: Yes

Reviewer #2: Yes

4. Is the manuscript presented in an intelligible fashion and written in standard English?

Reviewer #1: Yes

Reviewer #2: Yes

5. Review Comments to the Author

Reviewer #1: I am delighted to review the article entitled “Data-driven Modeling to Improve Agricultural Training Participation and Gender Inclusivity”. Over all this study have potential implications, but it needs to be improved in several parts, and I have noticed some typo errors throughout the manuscript. The survey methods and data used in this manuscript seems inadequate to be published in this high ranked journal, it still needs to be comprehensively explained with the clearer objectives. The analytical approach needs to be improved and shorten to specific points. I do appreciate that the topic may have potential connections to ethics literature, but unfortunately at present these areas are not well developed. Regarding contributions, at the end of the paper your discussion appears rather descriptive about what you have done in your study. As such, your paper appears rather limited in terms of addressing or elaborating a specific theoretical/ethics puzzle that would appeal to the audience/readers. an important topic. My detailed comments are as follows:

Abstract of this study needs to be rewritten with the actual theme of the study, policy recommendations should be provided. I would recommend to expand it with more information about the topic.

Authors are required to clearly explain the study objectives/research questions in the last paragraph of introduction part. What is the contribution of this paper? What are the novel points?

The theoretical framework is weakly described. The research hypothesis are not provided, it should be revised and presented in an articulated way with more updated citations. Further, it is recommended to provide the framework in picture/figure form (using MS. Visio).

Literature review section is missing, authors are required to add a “Literature review” section.

It is advised to revise the model settings, and parameters presented in this study.

The authors need to provide enough literature review to support your study hypothesis, literature in the table form would be more good.

I have noticed several typo errors throughout the manuscript, further this manuscript is poorly organized, more work need to be done on writing style and organization of the paper.

Reviewer #2: This paper studies an important subject, related to how we can promote gender inclusivity in agricultural trainings. It is demonstrating a data-driven approach to maximize participation. This could be a valuable contribution for practitioners involved in planning agricultural trainings, and I think the contribution would go beyond trainings on ICTs (which the authors may want to highlight), but I have a few concerns related to the paper.

First, attributes of a training such as the gender of the trainer or the time of the day during which it is organized, could be correlated with unobserved variables such as the workload that prospective participants might face at different times of the day - and thereby agricultural cropping patterns or responsibilities in a given location outside of the house, or the availability of female trainers in a given union or upazila. The variables that are entered in the model are not exogenous, and I am very concerned about omitted variable bias in this case. This warrants more discussion as a limitation in the manuscript, including a discussion on page 13 on how experimentation with some of these variables could give one the exogenous variation that one needs to really say something about the effects of a male versus female trainer, or the timing at which an event is organized. In general, interpreting the presented data and models as information on "training preferences" and to speak of "impacts" really seems like a stretch.

Second, and potentially related, it seems to me that there are crucial factors that might influence participation are excluded. For instance incentives (for instance transport money or whether people are provided with money versus meals) should be a key determinant of participation, and also, I would imagine that the way in which a training is introduced to participants, for instance whether men or women are recruiting the participants, or the communication channels through which the events are being announced, could greatly influence participation. Do the authors have any data on this that could be explored? That would enrich the model and potentially help address some of the concerns about unobserved confounding characteristics in my previous comment. Even without such data, more insights on how participants were recruited / invited to the training sessions seems important context that is lacking at the moment.

Third, whilst I understand that division is an important correlate of participation, it leads to somewhat weird conclusions. For instance, Rajshahi was selected frequently in the top female training list. Does this mean that we should organize more training sessions in Rajshahi, and ask women from other divisions to come there? Or simply ignore women in other divisions? I'm sure this is not what the authors have in mind. It might be good to predict attendance irrespective of divisions, so as a first step, look at what types of events would increase turnout and female participation across all divisions.

Fourth, the discussion on "reducing training variability from lists of 100 to 50 to 10" is unclear. I think this is related to the comment that one needs to balance machine learning with performance, and that overemphasizing early model performance may restrict learning, but this point could be clarified more.

Finally, given that the number of variables on page 18 is not that large, and most interaction terms appear being dropped using the variable reduction methods, I'm wondering what the added value is of using this data-driven approach as opposed to just estimating a Poisson count model in STATA or R, and indeed, it would be good to see those regressions for the reader to see how the different variables from page 18 are related to participation.

A minor comment: The sessions that were dropped were not different in size, at least not significantly so, from the sessions that were not dropped. This might be worth emphasizing in the manuscript.

6. PLOS authors have the option to publish the peer review history of their article (what does this mean?). If published, this will include your full peer review and any attached files.

Reviewer #1: No

Reviewer #2: No

---

## [Author Response · Author response to Decision Letter 0]

6 Oct 2022

Please see the point-by-point response to the reviewer and editor comments included in the resubmission.

---

## [Decision Letter · Decision Letter 1]

31 Oct 2022

PONE-D-22-16835R1Machine Supported Decision Making to Improve Agricultural Training Participation and Gender Inclusivity

PLOS ONE

Dear Dr. Reeves,

Thank you for submitting your manuscript to PLOS ONE. After careful consideration, we feel that it has merit but does not fully meet PLOS ONE’s publication criteria as it currently stands. Therefore, we invite you to submit a revised version of the manuscript that addresses the points raised during the review process.

The paper has partially improved following the comments of the reviewers, however there are still points to be addressed carefully in order to meet PLOS ONE publication criteria. 

Please take particular care to the comments raised by Reviewer 2(R2) .

Please explicitly provide clearer motivations on the use of a machine learning approach over simpler models.

Explain better the the used training split for the data and the other choices employed in this step.

Provide more proof about the generalization capability of the used methodology, providing a clearer validity of the overall approach.

Please take carefully into account all the comments for improving the manuscript to meet PLOS ONE standards before resubmitting it to the journal.

We look forward to receiving your revised manuscript.

Kind regards,

Sergio Consoli

Academic Editor

PLOS ONE

Reviewers' comments:

Reviewer's Responses to Questions

**Comments to the Author**

1. If the authors have adequately addressed your comments raised in a previous round of review and you feel that this manuscript is now acceptable for publication, you may indicate that here to bypass the “Comments to the Author” section, enter your conflict of interest statement in the “Confidential to Editor” section, and submit your "Accept" recommendation.

Reviewer #1: All comments have been addressed

Reviewer #2: (No Response)

2. Is the manuscript technically sound, and do the data support the conclusions?

Reviewer #1: Yes

Reviewer #2: No

3. Has the statistical analysis been performed appropriately and rigorously? 

Reviewer #1: Yes

Reviewer #2: No

4. Have the authors made all data underlying the findings in their manuscript fully available?

Reviewer #1: Yes

Reviewer #2: Yes

5. Is the manuscript presented in an intelligible fashion and written in standard English?

Reviewer #1: Yes

Reviewer #2: Yes

6. Review Comments to the Author

Reviewer #1: This manuscript can be accepted in current form, as the authors have well addressed all the concerns.

Reviewer #2: I thank the authors for considering my comments, and for their thoughtful response. I believe that the paper has improved in that it is more nuanced, and is improved in its readability.

At the same time, it seems like some of my comments are not fully addressed. I suggested additional analyses that were not done, without good reason for not doing so.

I am still concerned about the added value of the machine learning approach over and beyond a simple regression model. The regression was already estimated for another paper, but that is not a reason for not comparing the results from the machine learning model with the predictions by that regression model, and pointing out explicitly what is changed by the ML model.

Further, I am still concerned about the context-specificity, and overpredicting to a specific context, with awkward conclusions such as "policymakers can maximize turnout of female farmers by organizing trainings in such and such area". Policymakers would want a more generalizable model, which they can apply to a country as a whole, even if at the cost of lower accuracy. Showing how much accuracy is lost by having a slightly more generalizable model would be extremely valuable in demonstrating how to best employ this ML approach, and do a bit more to address my concern than just including a sentence or two discussing the limitation - I am worried that those few sentences are too easily missed.

Finally, the manuscript still speaks of training "preferences" when in fact there is too much omitted variable bias and too little control over the invitation process to be able to interpret turnout as "preferences". What is called "preferences" could simply be "presence" when the events are being marketed and a crowd is attracted; or "permission to go check out what's happening", or "time availability", correlated with the determinants going into the ML model, and I really think the word "preferences" should not be used here.

7. PLOS authors have the option to publish the peer review history of their article (what does this mean?). If published, this will include your full peer review and any attached files.

Reviewer #1: No

Reviewer #2: No

---

## [Author Response · Author response to Decision Letter 1]

29 Dec 2022

See Response to reviewer attachment.

---

## [Decision Letter · Decision Letter 2]

24 Jan 2023

Machine Supported Decision Making to Improve Agricultural Training Participation and Gender Inclusivity

PONE-D-22-16835R2

Dear Dr. Reeves,

We’re pleased to inform you that your manuscript has been judged scientifically suitable for publication and will be formally accepted for publication once it meets all outstanding technical requirements.

Kind regards,

Muhammad Khalid Bashir, PhD

Academic Editor

PLOS ONE

Additional Editor Comments (optional):

Reviewers' comments:

Reviewer's Responses to Questions

**Comments to the Author**

1. If the authors have adequately addressed your comments raised in a previous round of review and you feel that this manuscript is now acceptable for publication, you may indicate that here to bypass the “Comments to the Author” section, enter your conflict of interest statement in the “Confidential to Editor” section, and submit your "Accept" recommendation.

Reviewer #1: (No Response)

Reviewer #2: All comments have been addressed

2. Is the manuscript technically sound, and do the data support the conclusions?

Reviewer #1: Yes

Reviewer #2: Yes

3. Has the statistical analysis been performed appropriately and rigorously? 

Reviewer #1: Yes

Reviewer #2: Yes

4. Have the authors made all data underlying the findings in their manuscript fully available?

Reviewer #1: Yes

Reviewer #2: Yes

5. Is the manuscript presented in an intelligible fashion and written in standard English?

Reviewer #1: Yes

Reviewer #2: Yes

6. Review Comments to the Author

Reviewer #1: This manuscript can be accepted in current form, as the authors have well addressed all the concerns.

Reviewer #2: The authors have addressed my remaining concerns and I recommend publication of the manuscript in PlosONE.

7. PLOS authors have the option to publish the peer review history of their article (what does this mean?). If published, this will include your full peer review and any attached files.

Reviewer #1: No

Reviewer #2: No

---

## [Editor Report · Acceptance letter]

7 Feb 2023

PONE-D-22-16835R2 

Machine-supported decision-making to Improve Agricultural Training Participation and Gender Inclusivity 

Dear Dr. Reeves:

I'm pleased to inform you that your manuscript has been deemed suitable for publication in PLOS ONE. Congratulations! Your manuscript is now with our production department. 

Kind regards, 

on behalf of

Dr. Muhammad Khalid Bashir 

Academic Editor

PLOS ONE